# Unusual Findings of Human-Associated Four-Nucleated *Entamoeba* Species in Captive Wild Animals

**DOI:** 10.3390/ani15010090

**Published:** 2025-01-03

**Authors:** Lorena Esteban-Sánchez, Juan José García-Rodríguez, Francisco Ponce-Gordo

**Affiliations:** Department of Parasitology, Faculty of Pharmacy, Complutense University, Plaza Ramón y Cajal s/n, 28040 Madrid, Spain; lorees01@ucm.es (L.E.-S.); jjgarc01@ucm.es (J.J.G.-R.)

**Keywords:** giant anteater, mantled guereza, lar gibbon, mandrill, chimpanzee, common rhea, *Entamoeba dispar*, *Entamoeba hartmanni*, *Entamoeba nuttalli*, genetic analyses

## Abstract

Several species of the genus *Entamoeba* infect humans and other animals, primarily non-human primates. The number of morphological characters that can be used for their identification is very limited and none are conclusive, making genetic analyses crucial. The study of parasites found in captive animals, such as those in zoos, is very important both for the health of the animals themselves and for understanding the potential for transmission between animals and humans (caretakers and visitors). In this study, human-related *Entamoeba* species such as *Entamoeba dispar*, *Entamoeba hartmanni*, as well as *Entamoeba nuttalli*, incorrectly considered to be exclusive to non-human primates, have been identified in captive wild animals at a Spanish zoo (Zoo Aquarium of Madrid, located in Madrid city, Spain) through genetic analysis. The presence of *E. dispar* has been confirmed for the first time in birds (in greater rhea, *Rhea americana*) and in previously unreported mammalian hosts, including primates (mantled guereza, *Colobus guereza;* lar gibbon, *Hylobates lar*) and anteaters (*Myrmecophaga tridactyla*). *Entamoeba nuttalli* was detected only in mandrills (*Mandril sphinx*) while *E. hartmanni* was found in both mandrills and chimpanzees (*Pan troglodytes*). These findings expand our knowledge of the epidemiology of these parasitic species and emphasize the need for routine monitoring to prevent transmission.

## 1. Introduction

The study of human infections cannot be separated from that of animals with which humans come into contact, be their domestic pets or livestock, wild animals in their natural environments (through hunting or fishing), or those kept in captivity such as zoo animals. Captive wild animals often exhibit a higher prevalence of parasites [1], and their close contact with humans presents a significant zoonotic potential. Considering intestinal amoebae, the most important species infecting humans due to its pathogenicity is *Entamoeba histolytica* [2]; genetic analysis has revealed infections with this species in non-human primates (NHPs) [3,4]. Other morphologically similar species reported in NHPs are *Entamoeba nuttalli* [5,6,7,8,9], historically considered exclusive to NHPs [8,9], *Entamoeba dispar* [5,6,10,11,12,13,14,15,16], and *Entamoeba hartmanni* [17,18]. Besides primates, human-associated *E. histolytica*-like species have been confirmed to infect rodents after genetic analysis [19]. The fact that the same species infects different hosts raises the possibility of cross-transmission, both among animals and to or from humans, as there have been some reported cases of zoonotic transmission to or from zookeepers in several European countries and Egypt [20,21,22,23,24].

The genus *Entamoeba* encompasses a large number of different, recognized species inhabiting the intestine of vertebrate hosts. They have a direct life cycle, with a trophozoite stage dwelling in the intestine and a cyst as the resistance and infective stage. These species are traditionally classified into four groups based on the number of nuclei in their mature cysts: non-cyst-forming species (*Entamoeba gingivalis*-like group), species with one-nucleated cysts (*Entamoeba bovis*-like group), four-nucleated cysts (*E. histolytica*-like group), and eight-nucleated cysts (*Entamoeba coli*-like group) [25]. The four-nucleated cyst group is the most studied; it includes free-living and parasitic species infecting amphibians, reptiles, and mammals, among them the pathogenic species *Entamoeba invadens* in reptiles and *E. histolytica* in humans and NHPs. The species in the other groups are generally considered non-pathogenic [2]. Most species were described in the first half of the 20th century. In addition to the number of nuclei, other morphological characters used for species differentiation included cyst and nuclei size as well as nuclear characteristics [26]. This limited set of morphological traits hinders the differentiation of *Entamoeba* species; earlier researchers considered the host species (which may provide different, even unique physiological conditions for growth and development) and the geographical origin of the organism as complementary criteria for species validation [27]. No significant taxonomic debates occurred during the second half of the 20th century. However, in the 1990s and early 21st century, the advent of genetic analyses, primarily based on *Entamoeba* ribosomal gene sequences (particularly, the small subunit rRNA gene, SSU-rDNA), led to the characterization of some previously accepted species [28,29,30], the description of new species (i.e., *Entamoeba ecuadoriensis* [28] and *Entamoeba struthionis* [31], and the resurrection of others (e.g., *Entamoeba suis* [29]). Genetic variability was observed in some species [30], reopening taxonomic debates regarding criteria for species identification. Currently, increasing evidence indicates that identifications based on morphology, host specificity, or geographical origin of the isolate are insufficient for accurately identifying *Entamoeba* species; instead, such identifications must rely on genetic data [32]. Although over 60 species have been described in *Entamoeba* [27], genetic data are available for fewer than half of them. Even when genetic data enable differentiation, some researchers have proposed synonymies [29]. Additionally, substantial genetic diversity within some accepted species has led to the informal classification of isolates into ribosomal lineages rather than validating previously named species or describing new ones [33]. Presently, the true number of *Entamoeba* species remains undetermined, and genetic analysis is the only reliable method for unequivocally identifying species or subspecific variants [32].

Our group has been monitoring parasite infections in wild animals held in captivity at various Spanish zoological institutions over the last 10 years [34]. *Entamoeba* cysts have been commonly detected in samples from herbivorous and some omnivorous species, most of them belonging to the non-pathogenic one-nucleated or eight-nucleated cysts groups. However, in some cases from one of the zoos, four-nucleated cysts have been observed in unusual and/or previously unreported host species, or in hosts where published records were inconsistent or lacked objective data. In the present study, the identification of these isolates through genetic analysis is presented.

## 2. Materials and Methods

### 2.1. Study Location and Host Species

The Zoo Aquarium of Madrid is situated adjacent to a large urban park in the city of Madrid. It has an open layout, with animals distributed across five zones representing different continents. The enclosures are open-air and include covered areas for resting. The ground is mostly natural, with grass covering the majority of sections; only in some cases, concrete flooring is used. Animals are grouped within enclosures, either separated by species or in mixed-species groups, based on their compatibility. The enclosures are separated by various types of fences, water channels, or visitor pathways. In a few instances (primarily for carnivores and non-human primates), the animals are housed in semi-enclosed facilities, bordered by glass panels or metal fences, with mesh covering the top.

This study focused on the identification of four-nucleated cyst-forming *Entamoeba* species that were occasionally detected by microscopy at different times between 2009 and 2024, in samples from chimpanzees (*Pan troglodytes*), mantled guerezas (*Colobus guereza*), lar gibbons (*Hylobates lar*), mandrills (*Mandrillus sphinx*), giant anteaters (*Myrmecophaga tridactyla*), or greater rheas (*Rhea americana*) (Table 1). Chimpanzees were housed in separate enclosures delimited by glass panels and covered with nets. Lar gibbons that tested positive included two individuals under quarantine isolation. Mantled guerezas and mandrills were kept in separate enclosures with concrete floors, also delimited by glass panels. Giant anteaters and greater rheas were housed in open-air enclosures with natural soil covered by grass; these enclosures were bordered by metal fences located close to visitor pathways. None of the animals presented clinical signs associated with *Entamoeba* infections. The anteater that tested positive for *Entamoeba* experienced intermittent diarrheic episodes, although none occurred at the time of the findings; these episodes were likely caused by a trichomonadid flagellate [35].

### 2.2. Sample Collection and Processing

Fresh faecal materials were collected by zookeepers during early morning cleaning. Samples consisted of pooled faeces free from soil or plant debris, and were kept in clean, new plastic containers of adequate size. Only for the anteaters, faeces were collected from individual animals. Samples were transported to the laboratory within 1–3 h and were either processed immediately upon arrival or stored at 4 °C until processing (maximum delay: 24 h).

In the laboratory, the samples were processed using the sodium acetate-ether stool concentration technique [36]. Briefly, a small amount of each sample (2–3 g) was homogenised in 30 mL of aceto-acid buffer (an aqueous solution containing 1.5% sodium acetate and 0.36% acetic acid) in 50 mL conical tubes and filtered through a metal sieve to remove large debris. The filtered suspension was transferred to new 50 mL conical tubes, vortexed for 30 s with diethyl ether (1:1 ratio), and centrifuged at 1500 rpm for 2 min. The aqueous sediment (at the bottom of the tubes) was examined microscopically on temporary slides stained with Lugol’s iodine; in some cases, permanent trichromic and Clorazol-black stained slides were prepared from the original sample. Cysts were photographed and measured with an Olympus DP20 camera on an Olympus BX51 microscope (Olympus, Tokyo, Japan). Positive samples from anteater, colobus, and greater rhea were processed at the time of their collection (e.g., in 2009), confirming the presence of four-nucleated cysts, and were subsequently frozen and stored. Recently, with the detection of four-nucleated cysts in other host species, the previously frozen samples were further processed alongside the newly collected positive samples.

### 2.3. Genetic and Phylogenetic Analyses

For the genetic identification of *Entamoeba* four-nucleated cysts found in the samples, DNA was extracted using the Speedtools tissue DNA extraction Kit (Biotools B&M Labs S.A., Madrid, Spain) according to the protocol provided. The DNA was stored at −20 °C until further analyses were conducted. A fragment of about 700 bp in the 3′ end of the small subunit rRNA gene (SSU-rDNA) was PCR-amplified by using primers E12D (5′-AACACGGGAAAACTTACCAAGACCG-3′) and EF2 (5′-TGATCCTTCCGCAGGTTCAC-3′). The PCR reactions were performed by using the mixture from the kit PuReTaq^TM^ Ready-To-Go^TM^ PCR beads (Merck KGaA, Darmstadt, Germany) in a total volume of 25 µL, containing 2 µL of each primer solution and 5 µL of the DNA solution. The PCR was made on an Eppendorf Master Cycler Gradient (Eppendorf AG, Hamburg, Germany) programmed as follows: initial phase 10 min at 94 °C (initial denaturation), 30 cycles of 1 min at 94 °C, 1 min at 55 °C and 3 min at 72 °C, and a final cycle of 10 min at 72 °C (final extension). Positive or negative controls were not included in the PCR reactions because the samples were selected for being microscopy-positive for four-nucleated *Entamoeba* cysts, and the diagnosis was based on sequencing the PCR products rather than on the presence or absence of amplification bands.

PCR products were visualized in a 1% agarose gel stained with Pronasafe (Condalab, Torrejón de Ardoz, Spain) using a UV transilluminator (NuGenius Syngene, Cambridge, UK). The products were purified using a QIAquick^®^ PCR Purification Kit (Qiagen, Hilden, Germany) and sequenced in both senses in an AbiPrism 3730xl DNA Analyzer (Applied Biosystems—now ThermoFisher Scientific, Waltham, MA, USA) with the PCR primers, in the Sequencing Service of the Complutense University. Chromatograms were visualized using ChromasPro^®^ v 2.1.10 (Technelysium Pty Ltd., South Brisbane, Australia) and compared to the available sequences in GenBank using the BLASTN^®^ algorithm [37] implemented in the National Center for Biotechnology Information website (https://blast.ncbi.nlm.nih.gov/Blast.cgi; accessed on 21 April 2024).

The sequences obtained in this study were aligned with those of other four-nucleated cyst-forming *Entamoeba* species available in GenBank; when a large number of sequences from the same species were available, a subset was randomly selected. The sequence *Entamoeba polecki* AF149913 was used as the outgroup. The alignment included 93 nucleotide sequences; where necessary, the 5′ and 3′ ends were trimmed to adjust to the length of the sequences obtained in this study. The alignment was performed using ClustalX ver. 2.1 [38] and refined by considering the secondary structures of the SSU-rRNA sequences, using the model proposed by Alfonso et al. (2012) [39]. Sequence-structure editing and alignment were conducted using the 4SALE program [40]. The final dataset included 823 positions.

Phylogenetic trees were constructed using the Neighbor-Joining method in MEGA-X [41]. The Tamura 3-parameter model of nucleotide substitution was selected as the best model of nucleotide evolution based on the Akaike information criterion and the Bayesian information criterion calculated in MEGA-X. Bootstrap resampling was performed 1000 times to estimate branch support.

## 3. Results

In all animals that were at some point infected with four-nucleate cyst-forming *Entamoeba* species, the infections were not sustained for long periods. Cysts were observed in successive samplings over two weeks (in anteaters and rheas), four weeks (in primates), and up to seven months in the mandrill. In rare and mostly isolated occurrences, the cumulative prevalence was close to 0% in all cases except for the mandrill, where all samples from May 2024 to the present date tested positive.

One-, two-, and four-nucleated cysts measuring 15–27 µm were observed in greater rheas (Figure 1A), while mature four-nucleated cysts measuring 19–25 µm were observed in one anteater and in the primates (Figure 1B,C). Four-nucleated cysts measuring 13–15 µm were found in chimpanzees (Figure 1D), while cysts of two sizes, 12–14 and 27–29 µm in diameter, were observed in the mandrill. In all cases, the cysts were spherical and the nuclei presented the typical morphology with a homogeneous ring of peripheral chromatin and a central endosome.

The PCRs yielded single amplicons of the expected size. Sequencing and chromatogram analysis revealed single sequences corresponding to *E. dispar* in rhea, anteater, mantled guereza, and lar gibbon, and to *E. hartmanni* in chimpanzee (Table 2). These sequences were 100% identical to others available in GenBank, except the *E. dispar* sequence from anteater (which differed by one extra ‘C’ at position 564 of the sequence now obtained), and the *E. hartmanni* sequence from chimpanzee, which had two extra bases (‘G’ at position 8 and ‘T’ at position 48 of the new sequence). Mixed sequences were present in the chromatograms of the mandrill samples over time: a comparative analysis and manual editing of the chromatograms allowed us to obtain sequences 100% identical to *E. nuttalli* and *E. hartmanni* (Table 2; Appendix A). All sequences have been submitted to the GenBank/EMBL/DDBJ databases under accession numbers PQ389461-PQ389467.

In the phylogenetic trees, species infecting poikilothermic hosts and the free-living *Entamoeba marina* were positioned basally relative to those infecting mammals (Figure 2). *Entamoeba hartmanni* sequences formed a monophyletic branch subdivided into two sister sub-branches, positioned basally to other species infecting primates and humans. The isolates from the chimpanzee and mandrill were placed within the same *E. hartmanni* sub-clade. *Entamoeba dispar* sequences clustered in a single branch closely related to *E. histolytica* and *E. nuttalli*. These two species formed two closely proximal branches: one exclusively containing *E. histolytica* sequences, and the other containing a mix of sequences identified as *E. histolytica* or *E. nuttalli* (Figure 3). The sequence identified in the mandrill was placed within the branch containing the *E. nuttalli* sequences. A more detailed analysis of sequences identified as *E. histolytica* available in GenBank revealed that nearly 80 entries were most likely misidentified and correspond to *E. nuttalli* (Table 3, Appendix A).

## 4. Discussion

Using genetic analysis, we have documented for the first time the presence of human-associated *E. dispar* and *E. hartmanni* in lar gibbon and chimpanzee, respectively; confirmed the presence of *E. dispar* in anteaters, and, to the best of our knowledge, reported *E. dispar* for the first time in birds (greater rheas) and *E. nuttalli* in mandrills. In general, the infections occurred sporadically and were time-limited, representing isolated cases. However, this does not diminish their significance in demonstrating that the hosts reported here can serve as reservoirs for certain human-associated *Entamoeba* species. The situation in mandrills is different. In this host, the detection of *E. coli*-like cysts was previously common in the Zoo Aquarium [34]. This changed in June 2024, when four-nucleated cysts were detected. Since then, eight-nucleated cysts have not been observed, while four-nucleated cysts have been consistently detected in weekly follow-up analyses.

The present study is the first to provide genetic identification of the *Entamoeba* species infecting anteaters. The same individual that was once found infected with *E. dispar* at the Zoo Aquarium has not been found infected by *Entamoeba* again [34]. Additionally, only one of the four anteaters in the zoo facilities tested positive for *Entamoeba*, this suggests that these findings are exceptional. However, the lack of genetic data from the few studies in which *Entamoeba* cysts were found [44,45,46] makes it impossible to determine whether anteaters can be considered regular hosts of *Entamoeba*, including *E. dispar*. Diniz et al. (1995) and Coke et al. (2002) [44,45] indicated the presence of unidentified species in anteaters with digestive disorders, while Rojano et al. (2015) [46] reported their findings as *E. dispar*. Prior to these authors, there was no previous data to validate these findings, and these publications do not provide descriptions or images to support the identification of the genus or to confirm that it is compatible with *E. dispar*; indeed, Coke et al. (2002) [45] even questioned whether it was *Entamoeba* or *Acanthamoeba*. Although we do not dispute these identifications, there is no bibliographic or documentary support to substantiate them as *E. dispar* or any other species, and the most accurate approach would have been to report the findings as *Entamoeba* sp. In these circumstances, it cannot be confirmed that previous findings are equivalent to those now reported. As discussed below, specific identifications of *Entamoeba* species should rely on genetic data rather than on previous citations (if they even exist) in a given host.

Previous studies on rheas have identified one-nucleated and eight-nucleated *Entamoeba* cysts [33,47,48,49]; Stensvold et al. (2011) [33] have identified the one-nucleated species *E. polecki* and *E. struthionis* by genetic analysis (considered as ribosomal lineages by these authors). Our study is the first to report tetranucleated cysts in this avian species, which correspond to *E. dispar*, another human species. The presence of immature cysts (Figure 1) indicates that this is not a case of spurious parasitism. It is noteworthy that these birds can be infected by protists typically associated with mammals, including not only the aforementioned *Entamoeba* species but also ciliates such as *Balantioides coli* [50]; the reasons for this are unknown [51]. Since the early work of our research group [49], we have analysed a large number of samples from these birds in our laboratory [34,48] and have found very few *Entamoeba*-positive cases (none consisting of four-nucleated cysts). This suggests that rheas can be hosts of *E. dispar* but cannot be considered as important reservoirs for this parasite.

Related to this, it is noteworthy that the detection of *E. dispar* in all four hosts occurred within a short time frame (concurrently for colobus, anteaters, and rheas) and disappeared in subsequent samplings. We cannot explain this, as it could be due to several factors, including the potential suboptimal suitability of some of these hosts or the implementation of treatments and hygienic measures that have prevented reinfections. However, infections with *E. hartmanni* and *E. nuttalli* persisted over time in mandrills and occurred concurrently in chimpanzees.

The first detection of *E. dispar* coincided with the arrival of a new anteater from another zoological institution to the Zoo Aquarium, which exhibited intermittent diarrhoea [35]. The second detection occurred in quarantined lar gibbons. The potential for parasite introduction through animal transfers between zoos has been previously documented [52]. The positive anteater was housed adjacent to the rheas and in close proximity to the colobus and lar gibbons. In this case, it is possible that transmission occurred passively through caretakers or environmental factors (rain, wind, etc.). Lar gibbons were later housed with orangutans in an enclosure with natural soil covered by grass, delimited by glass panels and metal fences, and covered with nets; neither the gibbons nor the orangutans were found later positive with *Entamoeba* [34]. Regarding *E. hartmanni* and *E. nuttalli*, we cannot propose a specific entry point for the parasite. In addition to the previously mentioned possibility, there is the potential for contagion from surrounding fauna or humans (visitors and caretakers). Previous studies have demonstrated the zoonotic potential and transmission (sometimes bidirectional) between animals and caretakers of parasitic protists such as *Blastocystis* sp., *Giardia* sp., *E. nuttalli*, and *E. dispar* [21,22,24,53]. The fact that there is at least a small difference in the obtained sequences for *E. hartmanni*, and the very distant times at which *E. dispar* was detected, indicates that distinct introduction events have occurred.

In NHPs, *E. dispar* and *E. hartmanni* have been identified by genetic analysis in a large number of species [4,5,6,7,10,11,12,13,14,15,16,17,18,24,54]. Morphological differentiation between these species and other *Entamoeba* species in the *E. histolytica* group that infect humans (*E. dispar*, *E. hartmanni*, *Entamoeba moshkovskii,* and *Entamoeba bangladeshi*), and in some cases other primates, is impossible. Only *E. hartmanni* can be differentiated by its smaller size, while the other species have indistinguishable cyst sizes [55], thus their identification must rely on biochemical or genetic methods.

Before being recognized as a distinct species, *E. dispar* was initially considered a non-pathogenic variant of *E. histolytica*. This aligned with observations in the animals infected in this study, which exhibited no symptoms. However, in humans, it has been proposed that *E. dispar* can be involved in symptomatic gastrointestinal disorders, including abdominal pain [56,57,58]. Oliveira et al. (2015) [58] highlighted the ability of *E. dispar* to infect and produce amoebic abscesses only when dealing with xenic strains, revealing this as a crucial factor for pathogenicity. The reports of *E. dispar* in NHPs (by genetic analyses) have been described in asymptomatic animals [5,6,10,11,13,14,15,16]. In NHPs, the pathogenic species *E. nuttalli* has been identified in macaques [8,9,59] and chimpanzees [7], and it has been suggested that the identifications made as *E. histolytica* in NHPs by several authors [60,61,62,63] corresponded to *E. nuttalli* [8]. In fact, a detailed analysis of *E. histolytica* SSU rDNA sequences from different sources available in GenBank have shown that some of them corresponded to *E. nuttalli* (Table 2, Appendix A). This suggests that identifications as *E. histolytica* based on clinical and morphological criteria may have underestimated the prevalence of *E. nuttalli* (and perhaps *E. dispar*) and its pathogenic potential. In our study, the animals harbouring *E. nuttalli* did not present symptoms throughout the observed period, as has been reported by other authors [6].

Although there are highly host-specific parasite species, this does not appear to be the case for at least some of the species described within the genus *Entamoeba*. When a species is described in a particular host, it is logical and customary to identify new findings in the same host as that species; otherwise, we would be constantly questioning all previous results, hindering scientific progress. However, this approach has proven to be flawed in *Entamoeba*. Some authors have continued to classify or identify *Entamoeba* species based on their host species [33,64]. Nonetheless, there is increasing evidence that the same species (or lineage, to avoid taxonomic controversies) can infect very different hosts [29,33,64]. Under these circumstances, for *Entamoeba*, we suggest that identifications based solely on morphological criteria and bibliographic records should be reported as *Entamoeba* sp. or as species-compatible or species-like, and the specific name should only be assigned if genetic analyses have been conducted to allow unambiguous identification. Even in such cases, comparisons with existing data are essential to avoid errors as those demonstrated in the comparative analysis regarding sequences identified as *E. histolytica* available in GenBank. Some sequences were submitted before or around the time of the proposal of *E. nuttalli* as a distinct species (in 2007); however, most of the sequences were submitted long after this proposal (2014 or later), when the *E. nuttalli* sequence was already available for comparison. Although many of these sequences have yet to be published by their respective authors in scientific papers, and therefore cannot yet be verified, the data suggest that *E. nuttalli* may infect not only NHPs but also humans and possibly other animals. Unambiguous identification through genetic methods is essential, given the potential sanitary and epidemiological importance of the findings.

## 5. Conclusions

This study is limited to the description of microscopy-positive isolates of four-nucleated cyst-forming *Entamoeba* species; then, it cannot be discarded that other hosts harbouring subpatent infections were also infected. The study provides novel data regarding the host range of the human parasites *E. dispar* and *E. hartmanni* in animals, as well as the identification of *E. nuttalli* in mandrills. The finding of *E. dispar* typically considered a mammalian parasite, in avian hosts (rheas), underscores the inadequacy of host species as a sole criterion for identifying *Entamoeba* species, emphasizing the critical role of genetic data in accurate species identification. When genetic analysis that would undoubtedly allow for species identification is not conducted, identifications should be best proposed as sp. or species-like. Even when genetic data have been obtained, correct comparisons should be made to avoid misidentifications. The analysis of sequences assigned to *E. histolytica* of diverse origin (in some cases, human origin) deposited in public genomic databases has revealed some incorrect annotations when, in fact, they correspond to *E. nuttalli*. This result broadens the host range of this species, including humans. The data obtained highlight the risk of zoonotic transmission of *Entamoeba* species from captive wild animals. Consequently, regular monitoring of these animals is necessary to identify potential risks of pathogen transmission between them and humans.

## Figures and Tables

**Figure 1 animals-15-00090-f001:**
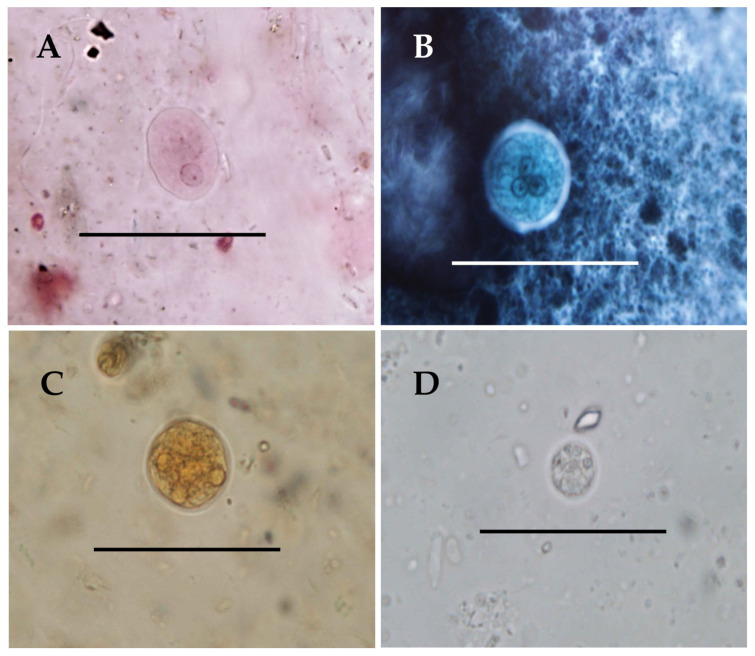
*Entamoeba* cysts found in the faeces of different hosts. (**A**) *Entamoeba dispar* from common rhea (*Rhea americana*), trichromic stain. (**B**) *Entamoeba dispar* from giant anteater (*Myrmecophaga tridactyla*), chlorazole black stain. (**C**) *Entamoeba nuttalli* from mandrill (*Mandrillus sphinx*), iodine stain. (**D**) *Entamoeba hartmanni* from chimpanzee (*Pan troglodytes*), unstained. Scale bar: 50 µm.

**Figure 2 animals-15-00090-f002:**
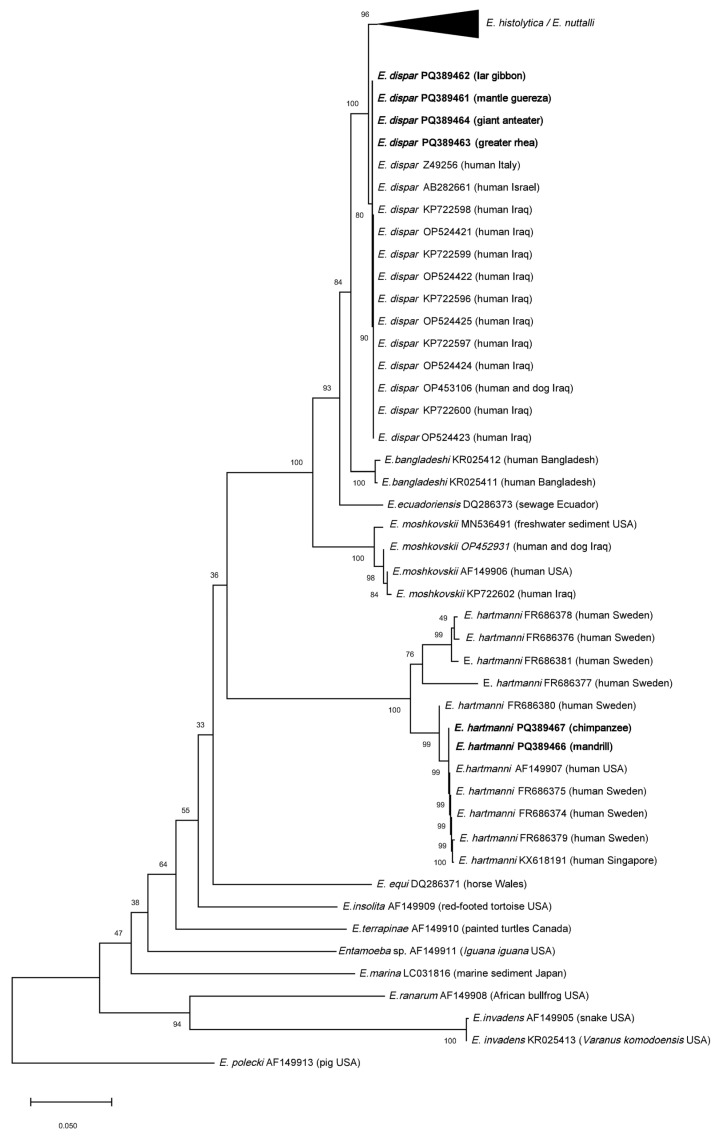
Phylogenetic relationships of the SSU-rRNA gene sequences of four-nucleated *Entamoeba* species by Neighbor-Joining method. The sequence *E. polecki* AF149913 is used as an outgroup. The number at the nodes represents the bootstrap support as computed from 1000 replicates. The tree is drawn to scale, with branch lengths measured in the number of base substitutions per site. The branches corresponding to *E. histolytica* and *E. nuttalli*, here collapsed, are shown in Figure 3. The sequences obtained in the present study are in bold.

**Figure 3 animals-15-00090-f003:**
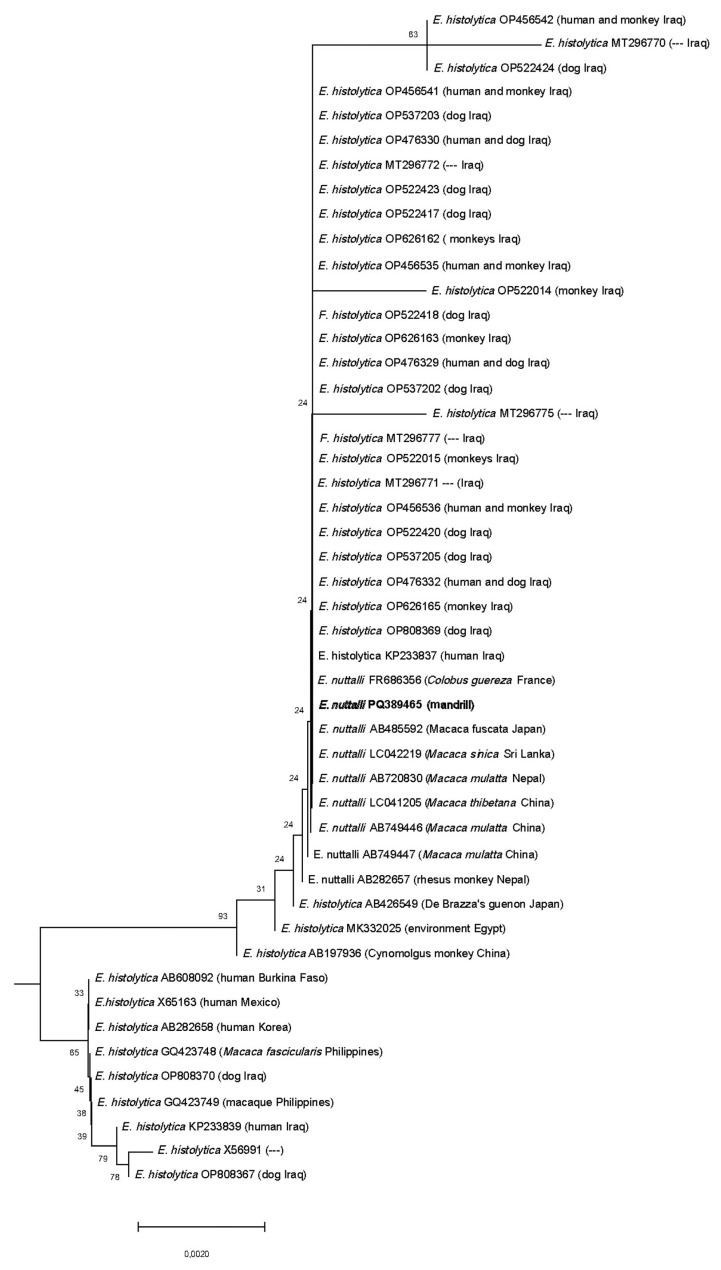
Subtree from Figure 2 presenting the phylogenetic relationships of the SSU-rRNA gene sequences of *E. histolytica* and *E. nuttalli* by the Neighbor-Joining method. The number at the nodes represents the bootstrap support as computed from 1000 replicates. The tree is drawn to scale, with branch lengths measured in the number of base substitutions per site. The sequence obtained in this study is in bold.

**Table 1 animals-15-00090-t001:** Characteristics of the samples found positive with four-nucleated *Entamoeba* cysts.

Host	Sampling Date(s)	Sample Characteristics
Chimpanzee	June–July 2024	Group of several individuals; infections by *Endolimax* sp. and *Balantioides coli* ^1^ regularly detected.
Lar gibbon	May 2023	Two animals under quarantine isolation. No other concurrent parasites were detected.
Mandrill	May–November 2024	Group of three individuals housed in two separate enclosures. In all cases, other regularly detected parasites included *Entamoeba chattoni* ^1^, *Enteromonas* sp., *Chilomastix* sp. and *Buxtonella* sp. ^1^
Mantle guereza	October 2009	Group of several individuals. *Trichuris* sp. Infections regularly detected.
Giant anteater	June 2009	Group of four individuals; one of which tested positive for *Entamoeba* and exhibited intermittent diarrhoea (not coinciding with positive samples). These animals were occasionally infected with *Tetratrichomonas* sp. [35], *Capillaria* sp. and *Balantioides coli* ^1^
Greater rhea	June 2009	Group of two individuals; no other concurrent parasites detected.

^1^ Confirmed by genetic analysis (Ponce-Gordo, F., unpublished results).

**Table 2 animals-15-00090-t002:** Genetic identification of the four-nucleated mature cyst-forming *Entamoeba* species found in this study.

Host	Species	Sequence Homology
Chimpanzee	*E. hartmanni*	99.70% AF149907
Lar gibbon	*E. dispar*	100% AB282661
Mandrill	*E. nuttalli*	100% LC041205
Mandrill	*E. hartmanni*	100% AF149907
Mantle guereza	*E. dispar*	100% AB282661
Giant anteater	*E. dispar*	99.86% AB282661
Greater rhea	*E. dispar*	100% AB282661

**Table 3 animals-15-00090-t003:** Misidentified *E. nuttalli* sequences available in GenBank. This is a selection of 8 sequences out of 78 in total (up to 21 April 2024) available in GenBank that are identified as *E. histolytica* but most likely correspond to *E. nuttalli*. Only sequences AB197936 [42] and AB426549 [43] (in bold) have been published; the rest were noted in GenBank as unpublished at the time of this writing.

Accession Number	Host	% Identity (BASTn) with *E. nuttalli* AB282657	% Identity (BASTn) with *E. histolytica* X65163
**AB197936**	*Macaca fascicularis*	1628/1630 (99.88%)	1617/1630 (99.20%)
**AB426549**	*Cercopithecus neglectus*	1628/1631 (99.82%)	1617/1631 (99.14%)
KP233837	humans	1628/1631 (99.82%)	1617/1631 (99.14%)
MW426070	sheep	753/757 (99.47%)	746/757 (98.55%)
OP522015	“Monkeys”	806/807 (99.88%)	799/807 (99.01%)
OP522425	dogs	804/807 (99.63%)	797/807 (98.76%)
OP526382	humans	455/455 (100%)	451/455 (99.12%)
OQ880537	humans	426/426 (100%)	422/426 (99.06%)

## Data Availability

The original contributions presented in the study are included in the article; further inquiries can be directed to the corresponding author.

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
