# Peer review of "Unusual Findings of Human-Associated Four-Nucleated Entamoeba Species in Captive Wild Animals"

_animals, 2025, doi:10.3390/ani15010090_

Round 1

Reviewer 1 Report

Comments and Suggestions for Authors

The manuscript by Esteban-Sánchez et al. is well-written and conducted, providing new and useful information on the epidemiology of Entamoeba species in captive wild animals. I support its further processing after appropriate modifications/clarification as outlined below:

Overall, I would like to advise the authors to present their research results under the Communication form (instead of a full-length article) and substantially increase the manuscript's word count. Generally, the MDPI journals request a minimum of 4000 words in the body of the main text.

L12: “we identify” – please avoid the personal mode verb formulation, as it may sound unprofessional in the scientific manuscript. Please revise this concern throughout the manuscript!

L38-48: In the “taxonomy paragraph“ of the introduction section, please ensure that the references used (from 1961 and 2011) provide up-to-date information. In my opinion, they must be replaced with recently published articles about the taxonomy topic of the Entamoeba genera.

L52: “… higher prevalence of parasites …” – replace with “frequency”. The “prevalence” term is more appropriate for the disease, and not for etiologic agents.

L54: “NHPs” – please define and avoid the direct use of acronyms

L59: the authors must provide data on the geographical spreading of zoonotic infections

L61: the authors must define the study location

L66-70: please reconsider the meaning of these sentences, in the present form seems to be results, rather than information related to the materials and methods section

L71: the sentence “…the morning” – please provide details (e.g. quantity, numbers, circumstances, etc.)

L81: “(SSU-rRNA)” – insert the “gene” word

L82: no available reference for the described PCR protocol

L83: “by using the kit” – unclear, do you want to say “mixture from the Kit”

L89: in order to validate their results, the authors must provide evidence of using positive and negative controls during PCR reactions. If not, this must be highlighted as a study limitation.

L92: “in in” delete “in”

L129: please insert a separate right column for Table 1 indicating the geographical origin of the mentioned sequences

L127: I would like to advise the authors to complete their study with a phylogenetic tree construction, this would significantly increase the interest of the scientific community

L215: within the conclusion section, the authors must highlight the study limitations and indicate further research directions in the approached research area

Author Response

Comment: The manuscript by Esteban-Sánchez et al. is well-written and conducted, providing new and useful information on the epidemiology of Entamoeba species in captive wild animals. I support its further processing after appropriate modifications/clarification as outlined below:

Response: Thanks very much for the time to read and comment the article, and for the overall opinion about it.

C: Overall, I would like to advise the authors to present their research results under the Communication form (instead of a full-length article) and substantially increase the manuscript's word count. Generally, the MDPI journals request a minimum of 4000 words in the body of the main text.

R: We have made many modifications to increase the word count, and now it is over the 4000 words limit. It was our mistake in the first version of the manuscript, to consider the word count utility of the Word processor – without noticing it included also title, authors, bibliography, etc.

C: L12: “we identify” – please avoid the personal mode verb formulation, as it may sound unprofessional in the scientific manuscript. Please revise this concern throughout the manuscript!

R: We have followed the comment and revised the manuscript accordingly. In our opinion, however, we think the use of the first person would not be unprofessional in scientific texts. Personally, I delight the reading of old papers (19th century – early 20th century), good studies, in which one can feel the problems and circumstances in which the authors did their research. An introduction such as that in Hoare (1940, Parasitology, 32: 226-237. doi:10.1017/S0031182000015717) is a good example of this.

C: L38-48: In the “taxonomy paragraph“ of the introduction section, please ensure that the references used (from 1961 and 2011) provide up-to-date information. In my opinion, they must be replaced with recently published articles about the taxonomy topic of the Entamoeba genera.

R: The taxonomy of the genus was a matter of research and debate in the first half of the 20th century. It entered in a stand-by period in the second half of the century until the development of genetic tools allowed go deeper in the characterization of species and comparisons between them. There was some debate in the 2000s decade, but after the proposal of Jacob et al. (J. Eukaryot. Microbiol. 2016, 63: 69-78), which personally I consider not well implemented, no further discussions have occurred. We have modified the introduction (paragraph #2) to provide a best description of the evolution of the taxonomy of the genus and its present situation, including the latest references about this topic.

C: L52: “… higher prevalence of parasites …” – replace with “frequency”. The “prevalence” term is more appropriate for the disease, and not for etiologic agents.

R: In this point, we disagree with the reviewer and we prefer to use the term “prevalence". According to Margolis et al. (J. Parasitol. 1982, 68, 131-133) and Bush et al. (J. Parasitol. 1997, 83, 575-583), prevalence is the number of hosts infected with one or more individuals of a particular parasite species (or taxonomic group) divided by the number of hosts examined for that parasite species. The term “prevalence” is of general use, not related only to disease. According to Bush et al. (1997: 577), “Frequency has been used when it is desirable to focus on the absolute number of infected hosts in a sample … rather than the proportion of the host sample they represent”.

C: L54: “NHPs” – please define and avoid the direct use of acronyms

R: Thank you very much for pointing out this error. It has been corrected and the acronym is spelled out when first used (lines 47-48). We think the use of this acronym in the text is justified, as we refer several times to non-human primates and we think it is easy to remember.

C: L59: the authors must provide data on the geographical spreading of zoonotic infections

R: Following the reviewer’s comment, we have added the geographical regions in which zoonotic transmissions between zoo animals and their zookeepers have been documented (line 54).

C: L61: the authors must define the study location

R: We have added subheadings in the material and methods section; the first subheading is “study location and host species”, and here we described the general characteristics of the ZooAquarium and the installations where the species found positive to four-nucleated Entamoeba cysts were housed (lines 100-109).

C: L66-70: please reconsider the meaning of these sentences, in the present form seems to be results, rather than information related to the materials and methods section

R: We have re-written these lines (lines 110-123 in the new version of the manuscript) in an attempt to clarify their meaning.

C: L71: the sentence “…the morning” – please provide details (e.g. quantity, numbers, circumstances, etc.)

R: We describe sampling in more detail in the second subheading of material and methods (2.2. Sample collection and processing, lines 128-133).

C: L81: “(SSU-rRNA)” – insert the “gene” word

R: We have made the correction as indicated (line 154). We have also modified the acronym as “small-subunit rRNA gene (SSU-rDNA)”, as the gene is DNA. We have revised the terminology in the text to maintain the consistency (SSU-rDNA when dealing with the gene, or the sequences, or SSU-rRNA when dealing with the secondary structure of the rRNA sequences).

C: L82: no available reference for the described PCR protocol

R: The protocol has not been copied from other papers; it has been designed by us, having into account the characteristics of the primers and the size of the fragment to amplify. It is not a special protocol, it can be considered as “normal” or “standard” (providing the annealing temperatures should be adapted to the Tm of the primers and the elongation time should be adapted to the length of the fragment to amplify).

C: L83: “by using the kit” – unclear, do you want to say “mixture from the Kit”

R: The text has been changed as suggested (line 156).

C: L89: in order to validate their results, the authors must provide evidence of using positive and negative controls during PCR reactions. If not, this must be highlighted as a study limitation.

R: Please note that the identification of one species or another was not based on the presence or absence of an amplification band but rather on the sequencing of the amplification products. Negative controls are useful to detect contamination in reagents or nonspecific DNA amplification; they ensure that the presence of amplified bands is solely due to the presence of parasite DNA. In this study, all samples were positive for four-nucleated Entamoeba cysts as confirmed by microscopic analysis. Consequently, amplification bands were expected in all cases. To verify that the bands corresponded to Entamoeba and not to nonspecific amplifications, sequencing was performed, rendering a negative control unnecessary.

Positive controls confirm that the primers and reaction conditions are suitable for amplifying the target parasite DNA. This ensures that the absence of amplification products is due to the lack of target DNA (i.e., an incorrect extraction process) or to suboptimal reaction conditions. However, in this study, amplification bands were consistently obtained from all microscopy-positive samples, as expected, and they can be used as positive controls. To further evaluate that the bands were not due to nonspecific amplification, sequencing was performed, making positive controls unnecessary as well.

Both positive and negative controls are optional in this context because sequencing serves as the quality control. This was clarified in the text (lines 162-165): “Positive or negative controls were not included in the PCR reactions, as the samples were selected for being microscopy-positive for four-nucleated Entamoeba cysts, and the diagnosis was based on the sequencing of the PCR products rather than the presence or absence of bands.”

C: L92: “in in” delete “in”

R: Deleted. Thanks very much for pointing out this error.

C: L129: please insert a separate right column for Table 1 indicating the geographical origin of the mentioned sequences

R: It was noted in the first version of the manuscript, and now it is stated in the subheading 2.1 of material and methods, that all animals were housed in the ZooAquarium of Madrid at the time of being infected by four-nucleated cyst-forming Entamoeba species. Then, the geographical origin of the sequences will be the same for all of them, the ZooAquarium in Madrid, Spain. As this is indicated in the material and methods, we consider it is not necessary to repeat it in the table.

C: L127: I would like to advise the authors to complete their study with a phylogenetic tree construction, this would significantly increase the interest of the scientific community

R: Following this comment, we have added a phylogenetic analysis. The material and methods, results and discussion sections have been modified accordingly.

C: L215: within the conclusion section, the authors must highlight the study limitations and indicate further research directions in the approached research area

R: We have modified the conclusion section according to the reviewer comment.

Reviewer 2 Report

Comments and Suggestions for Authors

The article is well written and edited! You can see the work put behind it! I would only have small observations to make that require remediation, of course this does not change the quality of the manuscript! I believe that this article deserves to be published due to its importance from a zoonotic point of view!

Please remove the underline from “ºC” from all the paragraph where you have it (most of them from Materials and Methods section)!

Please remove the space between the paragraphs from the whole article where necessary!

Line 140: Please add “,” after “Entamoeba bangla-140 deshi” from “(Entamoeba bangla-140 deshi. E. dispar, E. hartmanni, Entamoeba moshkovskii)”!

Line 150: Please remove Italics from “Oliveira et al. (2015) [32]”!

Line 168-191: Please remove Italics from “et al.” where necessary!

Author Response

Comment: The article is well written and edited! You can see the work put behind it! I would only have small observations to make that require remediation, of course this does not change the quality of the manuscript! I believe that this article deserves to be published due to its importance from a zoonotic point of view!

Response: Thanks very much for your comments and for the effort to read the manuscript. 

C: Please remove the underline from “ºC” from all the paragraph where you have it (most of them from Materials and Methods section)!

R: Done as requested.

C: Please remove the space between the paragraphs from the whole article where necessary!

R: We have revised the manuscript and deleted the extra lines between the paragraphs.

C: Line 140: Please add “,” after “Entamoeba bangla-140 deshi” from “(Entamoeba bangla-140 deshi. E. dispar, E. hartmanni, Entamoeba moshkovskii)”!

R: Thanks for pointing out that error, it has been corrected.

C: Line 150: Please remove Italics from “Oliveira et al. (2015) [32]”!

Line 168-191: Please remove Italics from “et al.” where necessary!

R: Thanks again for noting this error. We have revised the manuscript and corrected it where necessary.

Reviewer 3 Report

Comments and Suggestions for Authors

Dear authors,

The manuscript title is “Unusual Findings of Human-Associated Four-Nucleated Entamoeba Species in Captive Wild Animals” and it aims to assess the presence of different species of Entamoeba in several different animal species by molecular biology.

The topic falls within the aims and scope of the journal. It is very relevant in the context of One Health, it is well written and the results are original and well supported. Congratulations!

Some particular suggestions/comments will be done here:

-        Line 54 – please write in full first NHP (this reviewer believes it is Non-Human Primates, but it should be stated before)

-       Line 59 – here it is background but maybe in the discussion the authors may hypothesize also the human to animals’ transmission as animals are always seen as “the bad” and the inverse transmission may also occur in some cases

-       Line 140 – please add “E.” before histolytica as histolytica alone does not exist in terms of taxonomy

-       Lines 140, 141 – please be coherent and write always “E.” instead of Entamoeba. Delete the dot and add a coma between species

-       Line 211 – please write Giardia sp.

Author Response

Comment: The manuscript title is “Unusual Findings of Human-Associated Four-Nucleated Entamoeba Species in Captive Wild Animals” and it aims to assess the presence of different species of Entamoeba in several different animal species by molecular biology.

The topic falls within the aims and scope of the journal. It is very relevant in the context of One Health, it is well written and the results are original and well supported. Congratulations!

Some particular suggestions/comments will be done here:

Response: Thanks very much for the comments and time to read the manuscript.

C: - Line 54 – please write in full first NHP (this reviewer believes it is Non-Human Primates, but it should be stated before)

R: Thank you for noting this error; we have added the meaning of the acronym when first used (lines 47-48 in the new version of the manuscript).

C: - Line 59 – here it is background but maybe in the discussion the authors may hypothesize also the human to animals’ transmission as animals are always seen as “the bad” and the inverse transmission may also occur in some cases

R: we have modified the text in different parts to indicate that transmission can be from animals to humans and viceversa. In example, lines 36 (in abstract), lines 54-55 (in introduction), and lines 320-327 (in discussion).

C: - Line 140 – please add “E.” before histolytica as histolytica alone does not exist in terms of taxonomy

R: Done as requested.

C: - Lines 140, 141 – please be coherent and write always “E.” instead of Entamoeba. Delete the dot and add a coma between species

R: We have followed the rule that when a species is first mentioned, it should be spelled in full and abbreviate the genus in next cites (except when starting a sentence). We notice this can lead to sentences in which different species are spelled some in full, some abbreviated. We prefer to keep the use of this rule, but if the reviewer or the editor ask for changing it, we will do it.

C: - Line 211 – please write Giardia sp.

R: Done (writing “Giardia” in italics and “sp.” in the normal type).

Reviewer 4 Report

Comments and Suggestions for Authors

Author Response

Comment: Entamoeba is a widespread species which infect humans and other animals. So it is very important to identify the morphology and molecular of Entamoeba. This study found and identified several Entamoeba in some captive wild animals. This increases our knowledge of the epidemiology of these parasitic species and highlights the need for routine controls to prevent transmission. It is scientific significance in the MS, however, some problems still exist. The specific comments and suggestions are as follow.

Response: We wish to thank the reviewer for the time to read the manuscript and for his/her useful comments.

C: 1.there are *2649* words in the main text, it is necessary for the author to expand the number of words in the article to make the content more convincing.

R: The reviewer is right; it was our fault. We rely on the word count feature of MS Word, without noticing this would count all words (including title, authors, bibliography). The text has been revised and expanded and now it is 4900 words in length.

C: 2.Simple Summary. You are supposed to add the sampling site briefly.

R: We now indicate the full name and location of the zoo in which samples analyzed in this study were collected (line 15). A detailed description is provided in material and methods section (lines 101-109).

C: 3.Abstract. Abstract needs to be rewritten.

(1) The background description is a little too much, it is recommended to simplify.

(2) It is necessary to supply the material methods and results of the study.

(3) You should summarize a solid conclusion.

(4) Please add the significance at the end.

R: We have rewritten the abstract according to these comments. The background description has been simplified (lines 22-24), the material and methods and the results of the study have been indicated (lines 26-28), and we have summarized the conclusions and added the significance of the findings (lines 33-36).

C: 4.Introduction.

(1) It would be better to describe the relationship between the research site and Entamoeba.

R: We have revised and extensively rewritten the introduction. We now state the relationship between the research site and Entamoeba (lines 90-97).

C: (2) It is suggested to supply the importance of genetic analysis in the Entamoeba.

R: The taxonomic background of Entamoeba and the importance of performing genetic analysis for species identification in this genus is now explained in lines 56-89.

C: 5.Materials and Methods. This part is not organized well and must be rewritten.

(1) You are supposed to add some key information about sample collection, such as the number, age, sex and sampling date of various animals, and so on.

R: We have provided the info available about the characteristics of the samples and of sampled animals (lines 110-123, and table 1). Please note that we did not collected the samples, they were collected by zoo personnel which took the (part of the) feces that were accessible and non-contaminated by water, soil or grass; and that for several of the host species that were found infected, the population consisted on a group of animals making the identification sample-individual not always possible this leading to collect pooled samples. The intention of the present study is not to provide a detailed epidemiological study of the four-nucleated cyst-forming species  in each of the positive host species, but to indicate the unusual, even rare, presence of those Entamoeba species in those non-habitual hosts, noticing the possible risks that this possess for the transmission to humans (and also, the possibility for the animals to become infected by parasites transmitted to them from humans).

C: (2) Line 73. If you are following the method in Reference 26, you need to briefly state the steps.

R: Added as requested (lines 134-139).

C: (3) Line 81. Please add the concrete information about the using marker.

R: We are unsure about what information is requested. Both the marker and the gene fragment that is amplified and sequenced are specificized: “A fragment of about 700 bp in the 3’ end of the small subunit rRNA gene (SSU-rDNA)” (lines 80-81 in the former version of the manuscript, lines 153-154 in the new version). If the reviewer comment is related to the size of the amplicon, “about 700 bp”, please note that the Entamoeba species belonging to the different morphological groups have the SSU-rRNA gene of different length, from around 1800-1900 bp in the one-nucleated group (the exact value varies depending on the species), to around 1900 bp in the four-nucleated group and about 2100 bp in the eight-nucleated group. These differences in length are consequence of the presence of several indels, some of them within the boundaries of the amplified fragment. As the text in the manuscript is providing a general information (the primers amplify a fragment of the Entamoeba gene, any species), we only indicate the approximate size for the species under study (instead of “a fragment of 600-800 bp”).

C: (4) It is necessary to supply the methods about related sequences downloaded in the GenBank.

R: We now explain (lines 176-179) that other Entamoeba sequences available in GenBank were selected because they corresponded to four-nucleated cyst-forming species; a random selection was made in some cases, and E. polecki was used as outgroup.

C: (5) Why not construct phylogenetic trees to explore the phylogenetic relationships between different Entamoeba?

R: The phylogenetic analysis has been added as requested. Material and methods, results, and discussion sections have been modified accordingly.

C: 6.Results.

(1) Are all animals tested positive? If not, the infection rate should be stated.

We have added a mention to the prevalence of the detected four-nucleated Entamoeba cysts (Lines 195-197). Please note that in this study, we investigated rare, occasional or unusual findings; a systematic analysis of the animals were reported in another study (Esteban-Sánchez et al., Animals 2024, 14, 813).  In the present case, the title (“unusual findings”) gives an idea of the frequence with which the cysts were observed: the infection rate, frequency, or better, prevalence, is near 0%. The exception is the samples from mandrills; we have included the latest results (obtained in November 2024), and we indicate that all samples from this host are positive. We have modified the text where appropriate to include November 2024 as the last date in which we have found four-nucleated Entamoeba cysts.

C: (2) Why do different kinds of animals have different sampling times?

R: Please note that animals were sampled on a  routinary basis, but we only report here the specific identification of the amoeba found in a few samplings. As this happened for a short time in different moments in different host species, we indicate when they were detected. This study can be considered as a kind of several case reports combined in a single report.

C: (3) Line 105- Line 110. It would be better to describe the specific results with Figure 1A/1B/1C/1D.

R: The text has been revised and the descriptions have been made respect to each figure (lines 198-204).

C: (4) You are supposed to create a table showing details such as positive infection rates, whether there is a specific host and other related information.

R: As indicated in a previous response, samples were collected by zoo personnel and there is no precise data about how many individual animals were infected (except for lar gibbons, two individuals; and anteater, one individual). The info available is presented in a new Table 1.

C: (5) Table 1. The format of the three-line table is incorrect, please modify it.

R: The format was correct in the .docx file, but for some unknown reason, it is not in the .pdf file. We have checked now to ensure it is correct in the .pdf file.

C: (6) It is necessary to supply some partial results of PCR.

(7) Where is the result of chromatograms?

R: Please note that the identification of the species is not based on the presence/absence of PCR amplicons, but on their sequencing. As all analyzed samples were positive by microscopy, the expected result was to obtain a band of the adequate size (as it occurred; we now indicate this in the text, line 210). Showing the results of the PCR, say the bands, is not enough to identify the species, it would only indicate that Entamoeba DNA was present in the sample and the partial SSU-rRNA gene was amplified – but also a non-specific amplification of DNA from other organism could also yield the band. Then, the results of the PCR per se are not enough. The only method that clearly identify the organism is by sequencing the amplicon, and the results of the sequencing are the chromatograms showing the sequences. The chromatograms are the raw data from which the sequences are obtained, and these sequences have been submitted to GenBank.

We consider (and this is the normal practice) that it is not necessary to publish the chromatograms, as in most cases there are nothing to interpret, the peaks are clear and so the sequence; this (the sequence as the final result of the chromatogram analysis) is what it should be published. In the present study, and to clarify the procedure that allowed us to obtain some of the sequences from a mixed infection, we present a partial fragment of the chromatogram from the sequencing of that isolate along with the corresponding “traduced” sequences (Supplemental figure 1). This demonstrates the correctness of the comparative analysis and manual editing performed to obtain sequences 100% identical to E. nuttalli and E. hartmanni.

C: 7.Discussion.

(1) It would be better to put the Table 2 in the result or in the supplementary documents.

R: The table 2 (now Table 3) has been moved to results, and a supplementary file 2 has been added including the sequences available in GenBank that have been identified as E. histolytica but most likely correspond to E. nutalli.

C: (2) You could go deeper into why do birds get infected with Entamoeda.

R: We are unsure about if the comment refers to why birds can be infected with Entamoeba, or why the rheas in the zoo get infected by it. In the first option, we cannot explain the reason (simply, they are hosts of this parasite). In the second option, we cannot establish the precise circumstances that lead to the different host species to become infected, and only general comments can be made (lines lines 315-324).

C: (3) Line 211 - Line 213. It is not appropriate to end with this sentence, you should implement the ending to a sentence related to this study.

R: We have re-written the discussion and it ends with a sentence related to this study.

C: 8.Conclusions. It is recommended to synthesize to one paragraph.

R: Following the recommendation, conclusions are now synthesized to one paragraph.

C: 9.References.

(1) Please verify the capitalization of the first letter of each word according the requirements of the magazine.

(2) The scientific name of the parasite should be in italics.

The errors in the references are related to the software used to generate them. We have revised the reference section, once generated, to correct the errors.

Round 2

Reviewer 1 Report

Comments and Suggestions for Authors

The authors correctly acknowledged all of my raised concerns, Congratulations!

Reviewer 4 Report

Comments and Suggestions for Authors

The authors have addressed the comments and suggestions of this reviewer, and the revised manuscript is acceptable for publication.